# DATA BRITTLENESS ESTIMATION WITH SELF-SUPERVISED FEATURES

## ABSTRACT

To what extent are model predictions sensitive to modifications in training data? Data attribution approaches have served to answer this question. These approaches can be used for estimating data brittleness i.e., identifying which subset of training samples had the highest positive influence on a test sample. However, these methods come at a high computational cost, are memory intensive, and are hard to scale to large models or datasets. Current state-of-the-art approaches require an ensemble of as many as **300,000 models**. In this work, we focus on a computationally efficient baseline centered on estimating two types of data brittleness metrics. Our baseline approach uses the image features from a **single** pretrained self-supervised backbone. In contrast to data attribution approaches, our approach does not explicitly utilize model information and focuses on the data. Our results show this simple assumption works well, achieving competitive performance with state-of-the-art attribution approaches on CIFAR-10 and ImageNet, under limited computational and memory requirements. Our work serves as a simple baseline, showing that effective data brittleness estimates can be achieved based solely on knowledge of the training data.

## 1   INTRODUCTION

The effectiveness of a machine learning system's performance hinges on the quality, diversity, and relevance of the data it is trained on (Halevy et al., 2009; Sun et al., 2017). In various real-world machine learning systems, e.g. in healthcare or finance, we often ask questions like, "Which training samples influenced this prediction?" or "How sensitive is this model's prediction to changes in the training data?" Counterfactual insights enable us to assess the impact of hypothetical changes in the data distribution, which in turn helps us understand the basis of the model's decisions and how to change the decision in the event of an error. These questions motivate research on *data attribution* methods, focusing on understanding which data points most strongly influence a model's outputs.

In principle, data attribution can be done perfectly by a brute-force leave-$k$-out strategy; simply train the model from scratch many times, removing $k$ data points each time. The user can then examine the impact of each data point by examining how the corresponding ablated model differs from the original. Clearly, this procedure is intractable for any realistic problem as there are innumerable subsets, and training even a single machine learning model can be almost prohibitively expensive. The goal of data attribution research therefore is to approximate this gold standard metric as closely as possible while simultaneously using as little computation as possible. As such, the field of data attribution is all about trade-offs between accuracy, runtime, and memory.

We focus on estimating data brittleness metrics, a counterfactual estimation task for evaluating data attribution approaches. Concretely, for a test sample, we find the smallest subset of training data that when removed/mislabeled causes the model to misclassify. State-of-the-art data attribution approaches utilize logits (Ilyas et al., 2022), or gradient (Koh & Liang, 2017; Park et al., 2023) information from an ensemble of models to evaluate these metrics. These techniques require retraining models on different subsets of data and other compute or memory-intensive strategies for high efficacy (Ilyas et al., 2022; Feldman & Zhang, 2020; Koh & Liang, 2017; Park et al., 2023). Thus, attribution approaches quickly become intractable as datasets become larger (Basu et al., 2021; Park et al., 2023) and applications more realistic, such as attribution for large-language models (Grosse et al., 2023).

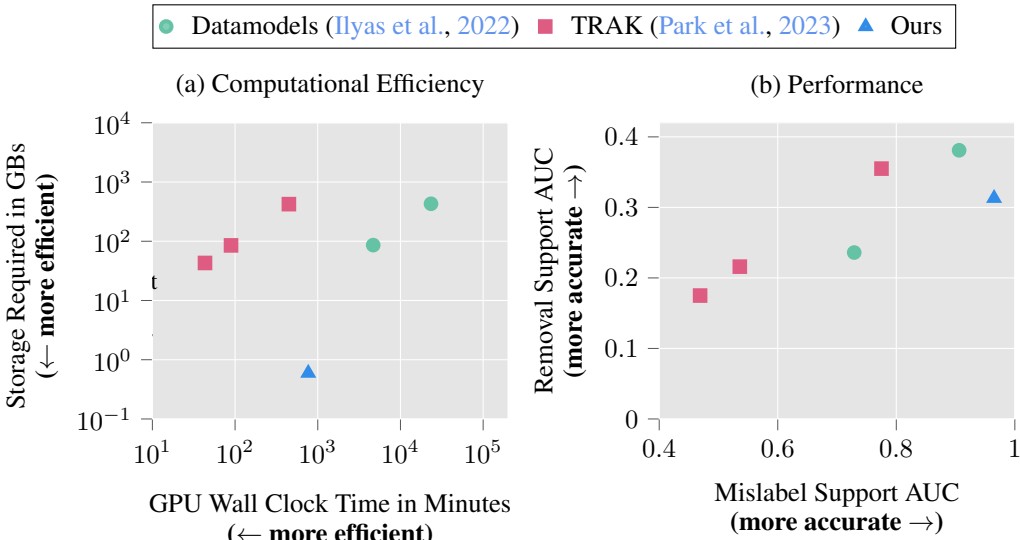

Figure 1: **Our proposed baseline approach for data brittleness estimation achieves high performance while improving computational efficiency**. Fig. (a) shows the wall-clock time on RTX A6000 GPU on the x-axis and memory requirements in GBs on the y-axis respectively (see Appendix A.1 for details). Figure (b) shows performance on two data brittleness metrics measuring the method's accuracy to make counterfactual predictions (details about the metrics are discussed in Section 2.1.)

Our baseline approach uses the feature space of a single self-supervised model to estimate data brittleness. We specifically focus on image classification. In contrast to data attribution approaches that focus on the specific way an algorithm/model behaves (Ilyas et al., 2022; Park et al., 2023), our approach does not explicitly utilize model information. Thus, our method cannot be directly utilized to study model-specific behaviors, considered applications of data attribution methods, like debugging model biases (Ilyas et al., 2022; Park et al., 2023; Shah et al., 2023), fairness assessment (Black & Fredrikson, 2021) or identifying backdoor attacks (Khaddaj et al., 2023). However, based on the simple intuition that different models leverage data similarly (Zhen et al., 2022; Kornblith et al., 2019), our approach can provide accurate brittleness estimates.

We show that our approach can outperform data brittleness estimates from state-of-the-art attribution approaches under limited compute and storage requirements on the CIFAR-10 dataset, as shown in fig. 1. Our method easily scales to much larger datasets such as ImageNet, while state-of-the-art attribution approaches require significantly higher compute or storage (see section 4.3). In contrast to claims in previous works, our results show that feature representations can serve as a simple, compute, and storage-efficient baseline for data brittleness metrics (Ilyas et al., 2022; Park et al., 2023). Our code is available as supplementary material.

## 2 BACKGROUND

We first define our notation and then discuss evaluation metrics used throughout the paper. These are borrowed from Ilyas et al. (2022) and Park et al. (2023).

**Notation**: Let $S = \{z_1, z_2, \ldots z_n\}$ denote a set of training samples. Each sample $z_i \in S$ represents $z_i = (x_i, y_i)$, where $x_i$ signifies the input image and $y_i$ represents the associated ground truth label. We use $z_t$ to denote an arbitrary evaluation sample not present in the training set. For a model trained on any training subset $S'$, with converged parameters $\theta^*$, we define a model output function on any sample $z$ as $f(z, \theta^*(S')) \in \mathbb{R}$. For the model output function, we use the correct-class margin (Ilyas et al., 2022):

$$f(z) = (\text{correct class logit}) - (\text{highest incorrect logit})$$

We denote a data attribution approach as a function $\tau(z, S) \in \mathbb{R}^n$. This function operates on any sample $z$ and a training set $S$, generating a score for each sample within the set $S$. These scores

highlight the relative positive or negative impact of individual training samples on the classification of the input sample $z$.

## 2.1 Evaluating Attribution Methods

A core evaluation criterion for the performance of data attribution methods is their capacity to provide accurate counterfactual predictions (Park et al., 2023; Ilyas et al., 2022; Feldman & Zhang, 2020; Koh et al., 2019). While these metrics can be computationally demanding, they represent a straightforward, yet valuable, proxy for assessing the efficacy of these approaches. In our work, we focus on one of the approaches presented in Ilyas *et al.*(Ilyas et al., 2022), and Park *et al.*(Park et al., 2023) and focus on **data brittleness**. Data brittleness metrics leverage attribution techniques to answer the following question: "*To what extent are model predictions sensitive to modifications in the training data?*" Hence, these metrics serve as a means of estimating counterfactual scenarios. To quantify data brittleness, we focus on two distinct types of data support for a test sample $z_t$. We define these sets below:

**Data Removal Support:** The smallest subset $R_r$, that when removed from the training set $S$, causes an average training run of the model to misclassify $z_t$.

**Data Mislabel Support:** The smallest training subset $R_m$, whose mislabeling causes an average training run of the model to misclassify $z_t$. For each training sample in $R_m$, we change the labels to the second-highest predicted class for $z_t$.

Intuitively, a better method should be able to find a smaller subset of training samples that can misclassify $z_t$. We estimate these metrics over a set of test samples and plot the cumulative distribution (CDF), which represents the probability that a sample's label can be flipped as a function of the data subset size. In fig. 1, we compare the Area Under Curve (AUC) of the CDF for the metrics described above across our approach and other attribution methods.

For a test sample $z_t$ and a data attribution approach $\tau(z, S)$, we rank the training samples based on decreasing order of positive influence on $z_t$. Then, based on the ranking, we iteratively select and modify a subset of training data. We perform this search, over different subsets to compute the smallest training subset that can cause $z_t$ to be misclassified. Naively, checking all possible subsets would be computationally expensive. (Ilyas et al., 2022) check only subsets with certain discrete sizes to keep costs manageable. We instead propose to perform a **bisection search** to approximate the search for the smallest subset, yielding more accurate results. The bisection search approximation is supported by the observation that several data attribution approaches are additive (Park et al., 2023). The exact algorithm and details are discussed in appendix A.6.

**Linear Datamodeling Score (LDS)** is another evaluation metric used for counterfactual evaluation of data attribution approaches (Ilyas et al., 2022; Park et al., 2023). Let $\{S_1, ..., S_m | S_i \subset S\}$ be $m$ random subsets of the training set $S$, each of size $\alpha \cdot n$ for some $\alpha \in (0, 1)$. The LDS metric is then defined as:

$$\text{LDS}(\tau, z) = \rho\bigg( f(z, \theta^*(S_j)) \mid j \in [m]\}, \{\tau(z, S) \cdot \mathbb{1}_{S_j} \mid j \in [m]\}\bigg)$$

where $\rho$ denotes Spearman rank correlation (Kokoska & Zwillinger, 2000), and $\mathbb{1}_{S_j}$ is the indicator vector of the subset $S_j$. In other words, LDS compares the Spearman rank correlation on a sample $z_t$ from $m$ models trained on different subsets of training data b/w ground truth class margin and predicted margin from a data attribution approach $\tau$. Note that while LDS metric accounts for both positive and negative influence in training samples, brittleness metrics only account for positive influence in training samples. Our baseline makes a simple assumption, that impacts performance on this metric.

Prior works focused on optimizing performance on LDS (Ilyas et al., 2022; Park et al., 2023), and evaluating data brittleness metrics as a downstream task. However, our results in section 5.1 indicate that LDS scores do not always correlate with brittleness estimates. We also note that LDS focuses on counterfactual predictions for *arbitrary* changes in training data, while data brittleness metrics serve to quantify counterfactual predictions using *targeted* changes to training data.

Other works have also evaluated data attribution methods using alternatives such as Shapley values or leave-one-out influences (Koh & Liang, 2017; Lundberg & Lee, 2017; Jia et al., 2021). These

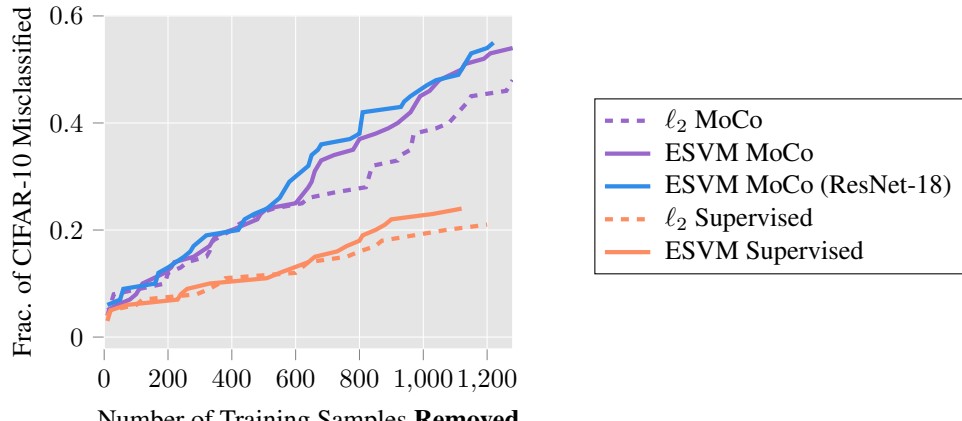

Figure 2: **Self-supervised features are more effective than supervised and are best compared using an ESVM**. Self-supervised features from MoCo can be used to find smaller data support than standard supervised features. For a larger fraction of test samples, ESVM distance is more effective than $\ell_2$ distance at ranking train images to select smaller data removal support.

approaches however are hard to scale beyond small datasets. An alternate line of work evaluates the utility of attribution methods for auxiliary tasks such as debugging model biases, active learning, or identifying mislabeled or poisoned data samples (Liu et al., 2021; Jia et al., 2021; Khaddaj et al., 2023; Shah et al., 2023).

## 2.2 BASELINES

**Datamodels** (Ilyas et al., 2022): In the *Datamodeling* framework, the end-to-end training and evaluation of deep neural networks is approximated with a parametric function. Surprisingly, the work shows optimizing a linear function is enough to predict model outputs reasonably well, given a training data subset. By collecting a large dataset of training data subsets and model output pairs, (Ilyas et al., 2022) demonstrate that such a linear mapping can accurately predict the correct class margin for individual test samples. Among other use-cases, these Datamodels are shown to be effective at counterfactual predictions and identifying visually similar train-test samples. But Datamodeling is prohibitively expensive, requiring the training of hundreds of thousands of models (300,000 in the original work) to accurately make counterfactual predictions. Unfortunately, this limitation makes Datamodeling intractable for all but small toy problems.

**TRAK** (Park et al., 2023): By approximating models with a kernel machine, *Tracing with the Randomly-projected After Kernels* (TRAK) makes progress toward reducing the computational cost of data attribution. This work uses a randomly-projected gradient information from an ensemble of models, to compute attribution scores. However, their method stores a high dimensional (from 4096 up to 20480 dimensional) projected gradient for each training and test sample, from a dozen or more model checkpoints to use as a "feature", leading to significant storage requirements. For our experiments, the total storage cost of using TRAK surpassed 400 GBs when using 100 models on CIFAR-10. Note that while our approach also stores feature embedding for each training sample, it only uses the penultimate layer of **single** self-supervised model, and thus our embeddings are much lower dimensional (128 dimensional for CIFAR-10).

## 3 OUR APPROACH

Our approach utilizes the penultimate feature space representation of a network to extract features from a test sample $z_t$ and each training sample in $S$. We approximate attribution scores by measuring the distance in feature space between $z_t$ and each training sample in $S$. Datamodels and TRAK have tried using features from an ensemble of supervised models and claimed them to be ineffective for counterfactual estimation (Park et al., 2023; Ilyas et al., 2022). Next, we describe the details of our approach.

**Feature extractor.** We find that the learning paradigm used for feature extraction heavily influences the estimation of data support. For example, embeddings from a ResNet-9 trained using a self-supervised learning objective (MoCo, (He et al., 2020)) can be used to find smaller support sets than the same model trained in a supervised manner (See $\ell_2$ MoCo vs $\ell_2$ Supervised in fig. 2). With the exception of DINO (Caron et al., 2021) (whose test accuracy on CIFAR-10 was much lower), all self-supervised feature extractors perform better than their supervised counterpart (see appendix A.3 for evaluation using multiple different feature extractors). We select MoCo as our preferred feature extractor, as it outperforms other self-supervised approaches in both data removal support and mis-labeling support scenarios. We find that the ResNet-18 backbone provides better support estimates than ResNet-9, and hence use it as default for all our experiments. Datamodels and TRAK, only compared against supervised feature extractors, leading to significantly worse counterfactual estimation.

**Distance function** When measuring the distance between two embeddings, Euclidean distance ($\ell_2$) is a common choice (Ilyas et al., 2022; Park et al., 2023). Cosine distance and Mahalanobis distance have also been used to measure similarity, but these were found to perform similarly to Euclidean distances in previous work (Hanawa et al., 2021; Ilyas et al., 2022; Park et al., 2023).

However, we find that measuring distance as distance to the hyperplane of an Exemplar SVM (ESVM) improves image similarity (Malisiewicz et al., 2011). To compute this, we train a linear SVM using only the target sample embedding, as a positive sample and treating all other samples as negative samples. We use the inverse of decision boundary distance, to measure the magnitude of influence. Hence, samples closest to the decision boundary have a higher influence and vice-versa. In fig. 2, we show using hyperplane distance to ESVM yields better removal support estimates than $\ell_2$ distance.

**Sign Estimation** Directly using $l_2$ or ESVM distance function, cannot identify training samples with negative influence. Intuitively, to change the prediction of a test sample, we should only remove samples from its vicinity that share the same class. While this assumption is simplistic, we show that it works well for estimating data brittleness. For a test sample, $z_t = (x, y)$ we simply assign all training samples with label $y$, as a positive influence, and negative otherwise i.e. given a target image of an airplane, only airplane training images have a positive influence. See appendix A.4 for more discussion on this.

## 4 Data Brittleness Estimation

We evaluate our approach on popular classification datasets i.e. CIFAR-10 and ImageNet, which are small enough to allow for some comparison with the more expensive approaches of TRAK and Datamodels.

### 4.1 Experimental Setup

**Training Setup:** We estimate the approximate data removal and data mislabel support for CIFAR-10 and ImageNet. As computing the data support for even a single validation sample requires training multiple models, we restrict ourselves to a reasonably small set of validation samples. We use the same validation samples across all methods. To accelerate the training of these models, we use the FFCV library (Leclerc et al., 2023). We use a similar setup to TRAK (Park et al., 2023). Our model training setup for evaluation is described in detail in Appendix A.2

**Baselines and Our Setup:** To estimate TRAK scores on CIFAR-10, we train 100 ResNet-9 models and use a projection dimension of 20480. To estimate scores on ImageNet, we train 4 ResNet-18 models and use a projection dimension of 4096. Computing TRAK scores using 4 models already requires 160 GB of storage space, hence we refrain from using a larger ensemble of models.

For Datamodels, we download the pre-trained weights optimized using outputs from 300K ResNet-9 models with 50% random subsets.[1] We also download the binary masks and margins to train our own Datamodels on outputs from 10K and 50K ResNet-9 models, using another 10K models for validation. Since Datamodels are extremely compute-intensive and require training hundreds of thousands of models, we cannot include them as a baseline on ImageNet.

---

[1]https://github.com/MadryLab/datamodels-data

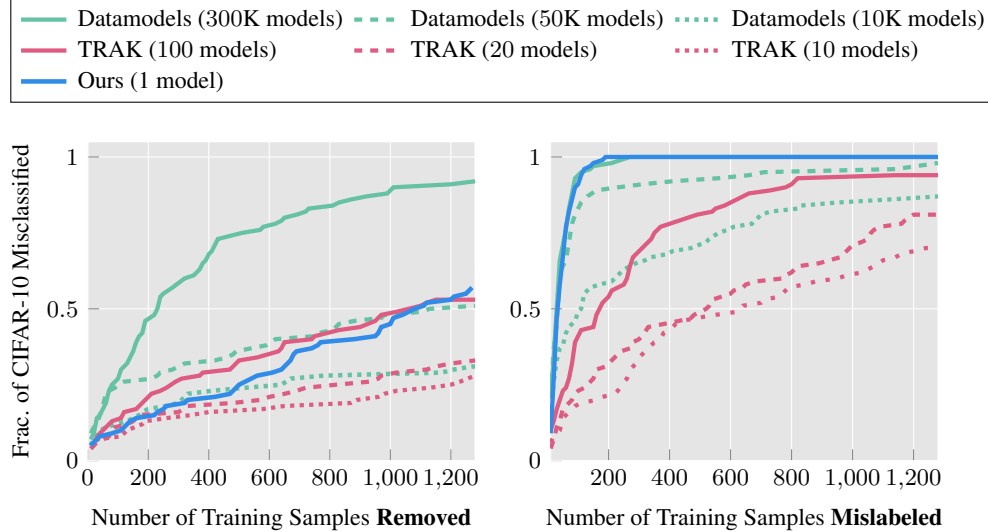

Figure 3: **Our baseline approach uses only a single model and outperforms TRAK and Data-models using 20 and 10,000 models for data brittleness metrics.** We estimate data removal and data mislabel support for 100 random CIFAR-10 test samples using a ResNet-9 model and plot the cumulative distribution using our approach and other baselines. The number of models used by each approach is also shown. For data removal support, using only a single model our proposed approach outperforms TRAK (Park et al., 2023) using 20 models and Datamodels (Ilyas et al., 2022) using 10,000 models. For data mislabel support, we outperform TRAK using 100 models and perform equivalent to Datamodels using 300,000 models.

For our baseline approach to train self-supervised models, we use the Lightly library (Susmelj et al., 2020). We train a ResNet-18 model using MoCo (He et al., 2020) for 800 epochs on CIFAR-10, using the Lightly benchmark code.[2] On ImageNet, we download a pre-trained ResNet-50 model trained using MoCo.[3] For our approach, we always use a single model. We denote Datamodels using N models as Datamodels(N), and similarly for TRAK.

## 4.2 CIFAR-10

In fig. 3, we present the distribution of estimated data removal values for CIFAR-10. Our findings reveal that employing a single model with a MoCo backbone (He et al., 2020) for data removal support proves more effective than employing Datamodels with 10K models and TRAK with 20 models. Our approach and Datamodels (10K) identify that 23% samples can be misclassified by removing fewer than 500 (example-specific) training samples while TRAK (20) can only identify 16%. For support sizes up to 1280 images, our approach identifies 55% of validation samples, whereas TRAK (20) and Datamodels (10K) can only identify 28% and 31% samples respectively.

In the same figure, we also depict the distribution of estimated data mislabel support for CIFAR-10. Here, our approach outperforms TRAK (100) and approaches the performance of Datamodels (300K). Here, our approach identifies 47% of CIFAR-10 validation samples that can be misclassified by mislabeling less than 30 training samples! In contrast, TRAK (100) performs poorly identifying only 20% of these samples. DataModels (300K) can identify 50% of validation samples marginally surpassing our performance.

In fig. 4, we further inspect how well our baseline approach works for each validation sample. We compare the individual estimated support sizes for all 100 samples using our approach versus other baselines. Our results show that for data removal support, across 16% of validation samples, our estimated data removal support is smaller than those of Datamodels (50K). For 44% of the samples our data removal estimates match TRAK and Datamodels (50K). For data mislabel support, our

---

[2]https://docs.lightly.ai/self-supervised-learning/getting_started/benchmarks.html
[3]https://github.com/facebookresearch/moco

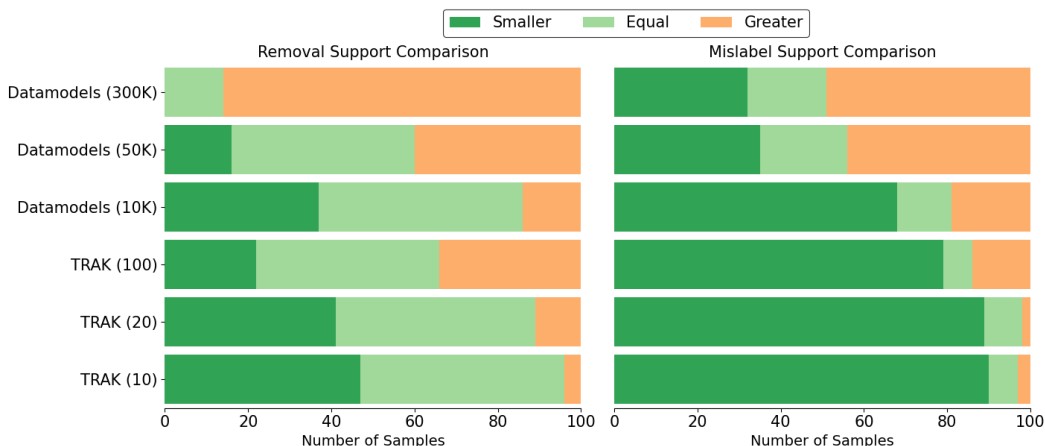

Figure 4: Compared to instances of Datamodels and TRAK, we check whether our data support estimates are smaller, equal, or larger for all 100 validation samples. For 32 samples, our proposed method can find smaller data mislabel support compared to Datamodels (300k models). Even for the data removal case, our approach finds an equivalent support estimate as Datamodels (300k models) for 14 samples.

approach finds a smaller support estimate than Datamodels and TRAK for 32% and 79% of the validation samples.

While our baseline approach cannot outperform Datamodels (300K) on data removal, our performance on the data mislabel support is nearly the same. Thus, our baseline approach of using a single self-supervised model can serve as a simple, compute, and storage-efficient alternative to estimate data brittleness.

### 4.3 IMAGENET

In fig. 6, we show our results for data removal on ImageNet. Our results show that much more accurate data brittleness estimates on ImageNet. In contrast, TRAK (1) and TRAK (4) do not scale well to ImageNet at all and provide much looser data removal estimates. We again emphasize that even scaling to TRAK with 10 models would require around 400 GB of storage space, by our estimate. Datamodels would require training tens of thousands of models on ImageNet, hence we cannot include it as a baseline. This highlights the scalability of our baseline approach where, under a limited compute and storage budget, a single self-supervised MoCo backbone can provide more accurate data removal estimates than existing data attribution methods.

### 4.4 TRANSFER TO A DIFFERENT ARCHITECTURE?

Datamodels and TRAK utilize information tied to the model architecture, such as gradients or logits from an ensemble of models. Being data attribution methods, Datamodels and TRAK approximate how training data influences *a particular model's* output. Thus, it is not appropriate to use a Datamodel for a ResNet-9 to attribute training data for a MobileNetV2. However, what could we learn if we tried to do so?

Different neural network architectures are known to exploit similar biases and output similar predictions (Mania et al., 2019; Toneva et al., 2018). In order to better understand the role of architecture in shaping these biases, we test how well attribution scores from these approaches transfer to other architectures for estimating data brittleness.

In fig. 5, we compare TRAK, Datamodels, and our attribution scores and evaluate them on a MobileNetV2 architecture (Sandler et al., 2018). We also show results across other diverse architectures in appendix B. As mentioned before, architecture, data augmentation, optimizer, etc. *must* play roles of varying importance in data attribution. But by changing the target model architecture, this experiment allows us to isolate and measure the importance of architecture. Comparing fig. 5 to fig. 3, we

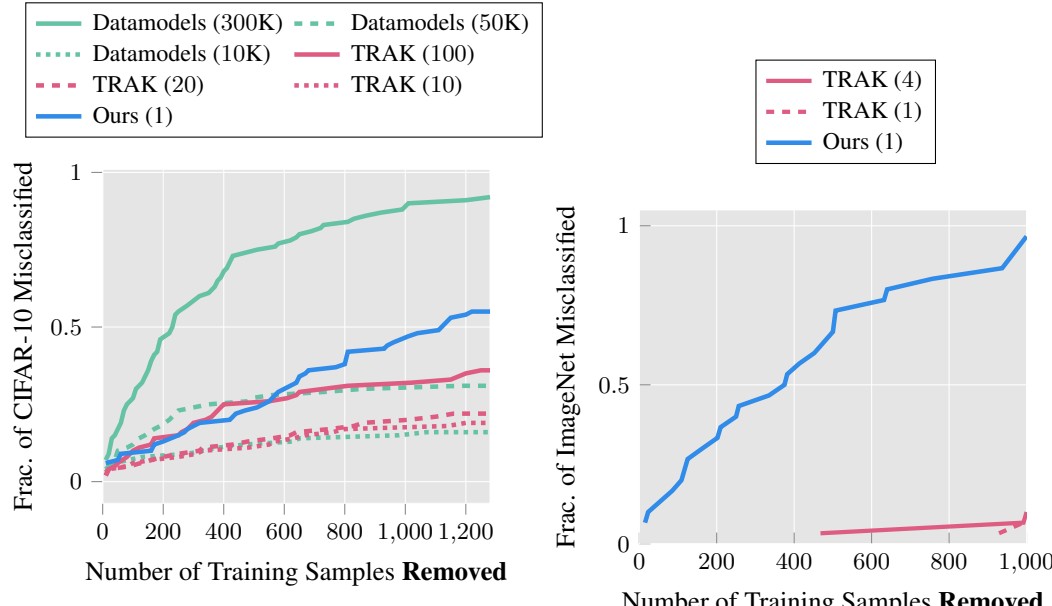

Figure 5: **The role of data is large on the counterfactual task of removal support.** We evaluate how attribution scores transfer from one architecture transfer to another. We use ResNet-9 scores for TRAK and DataModels and estimate data removal support for MobileNetV2. For our approach, we use the same ResNet-18 backbone.

Figure 6: **Our method yields better upper bounds on support size compared to TRAK-4, which requires more storage than the ImageNet dataset itself.** We estimate data removal support for 30 random ImageNet validation samples and plot the CDF of estimates. We use Resnet-18 for our data removal esimates.

find that DataModels (300K) remains surprisingly effective at estimating small support sets, suggesting that attribution scores for this method could be relying more on non-architectural factors of the training pipeline. By design, our approach using ResNet-18 predicts accurate data removal estimates surpassing TRAK (100) and Datamodels (50K). Datamodels with under 50K models and all TRAK variants decrease in their ability to estimate removal support, suggesting these methods are slightly more reliant on architectural information than DataModels (300K).

Our baseline approach using ResNet-18 can also be viewed as an ablation on all non-data factors, given that only data is common between the ResNet-9 training pipeline used in Datamodels and the MoCo self-supervised training pipeline we use. From this perspective, the strong performance of our baseline suggests the role of data is large on this counterfactual task of removal support. Our results also suggest that a much simpler prior can be leveraged to achieve effective data brittleness estimates.

This experiment also opens new avenues of research focused on understanding precisely how attribution methods score training samples. For instance, one could ablate each factor of the training pipeline and precisely measure how Datamodels or TRAK is utilizing information about training augmentations, optimizer, etc. Since our focus is data brittleness estimation, these directions fall outside our current scope and are deferred for future research.

## 5 DISCUSSION

### 5.1 LINEAR DATAMODELING SCORE

For computing the LDS score, we slightly adapt our baseline approach setting scores beyond the highest top-5% in magnitude to be zero, leading to sparser attribution scores. The sparsity prior has been shown to be effective in TRAK and Datamodels. In table 1, we compare LDS scores using our baseline approach, TRAK, and Datamodels. Datamodels framework uses tens of thousands of

| | Models Used | LDS Scores |
|---|---|---|
| | 300,000 | 0.56 |
| Datamodels | 50,000 | 0.43 |
| | 10,000 | 0.24 |
| | 100 | 0.22 |
| TRAK | 20 | 0.15 |
| | 10 | 0.12 |
| | 5 | 0.08 |
| Ours | 1 | 0.08 |

Table 1: We compare LDS scores for our approach with other baselines on CIFAR-10. Our proposed approach can perform equivalent to TRAK with 5 models.

models, to optimize for predicting the correct class margin. Hence, it achieves high LDS. TRAK also does well with multiple models. Our baseline shows a relatively low correlation on LDS, equivalent to TRAK using 5 models. The low correlation with LDS implies that these approaches may not be estimating training samples with negative or zero influences well, due to our simplistic assumption of using all samples of non-ground truth class as negative influences.

A lot of emphasis has been placed on the LDS metric by SOTA attribution methods while it is meant to serve only as a simple and efficient proxy for evaluating counterfactual scenarios (Ilyas et al., 2022; Park et al., 2023). However, our results show that low LDS scores do not necessarily imply worse performance on other related counterfactual estimation tasks and suggest that attribution scores for LDS, and data brittleness estimation may be less correlated than shown in prior works.

## 5.2 ROLE OF VISUAL SIMILARITY

In fig. 7, we plot the most similar training images according to Datamodels, TRAK, and our method. Given that our approach relies on comparing MoCo features from the same class as the target image, it makes sense that the closest training images are visually similar. On the other hand, the most similar training images found by Datamodels (Ilyas et al., 2022) and TRAK (Park et al., 2023) show more variability.

Despite the variability of most similar train images, Datamodels (300K) outperforms all other methods in the counterfactual estimation tasks, underscoring the importance of modeling additional non-data factors. Still, our method shows the significance of relying on visually similar training samples, implying that strong performance can be achieved for data britleness estimation, with much less explicit knowledge of the learning algorithm.

## 6 OTHER RELATED WORKS

Data attribution methods should produce accurate counterfactual predictions about model outputs. Although a counterfactual can be addressed by retraining the model, employing this straightforward approach becomes impractical when dealing with large models and extensive datasets. To address this problem, data attribution methods perform various approximations.

The seminal work on data attribution of Koh & Liang (2017) proposes attribution via approximate *influence functions*. More specifically, Koh & Liang (2017) identify training samples most responsible for a given prediction by estimating the effect of removing or slightly modifying a single training sample. But being a first-order approximation, influence function estimates can vary wildly with changes to network architecture and training regularization (Basu et al., 2021). Nevertheless, approximations of influence functions have also been attempted for multi-billion parameter models (Grosse et al., 2023).

Measuring empirical influence has also been attempted through the construction of subsets of training data that include/exclude the target sample (Feldman & Zhang, 2020). In a related approach, TracIn (Pruthi et al., 2020) and Gradient Aggregated Similarity (GAS) (Hammoudeh & Lowd, 2022a;b) estimate the influence of each sample in training set $S$ on the test example $z_t$ by mea-

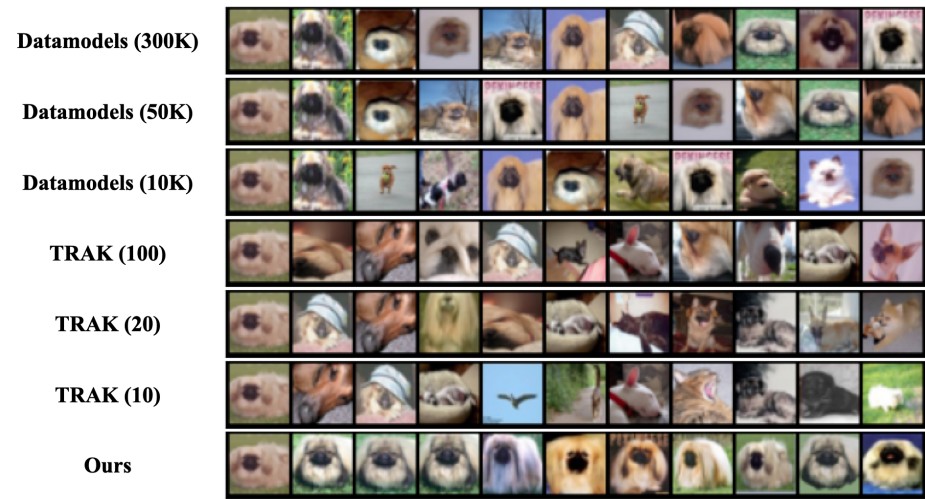

**Datamodels (300K)**

**Datamodels (50K)**

**Datamodels (10K)**

**TRAK (100)**

**TRAK (20)**

**TRAK (10)**

**Ours**

Figure 7: **Our attribution method consistently selects the most visually similar training images by design**. In each row, we plot the same target test image (Index 31), followed by ten most similar training images according to each attribution method.

suring the change in loss on $z_t$ from gradient updates of mini-batches. Another related line of work has used Shapley values to ascribe value to data, but since Shapley values often require exponential time to compute, approximations have been proposed (Ghorbani & Zou, 2019; Jia et al., 2019). In general, there seems to be a recurring tradeoff: computationally efficient methods tend to be less reliable, whereas sampling-based approaches are more effective but require a large number of models.

## 7   LIMITATIONS

We emphasize that our approach is not meant to serve as a traditional data attribution method since it only uses information regarding the training data. Our approach relies on the intuition that models leverage data in similar ways (Zhen et al., 2022; Kornblith et al., 2019). It doesn't explicitly utilize information about model architecture, optimizer, or other aspects of training strategy. Since it relies on visual similarity it may not directly perform well for certain downstream tasks such as debugging biases specific to the training algorithm (Shah et al., 2023) or identifying backdoors (Khaddaj et al., 2023). Also, unlike TRAK and Datamodels, it is unclear if utilizing a large ensemble of models would benefit performance on our evaluation metrics.

## 8   CONCLUSION

Data attribution approaches are computationally expensive, require an ensemble of models, and can be prone to inaccuracy. While these approaches exhibit promise and capability, their scalability to large-scale models remains uncertain. In this work, we present a simpler approach using features from a single self-supervised model, and evaluate it on two different counterfactual estimation scenarios: data removal support and data mislabel support. Our approach scales easily and provides tractable brittleness estimates on larger datasets such as ImageNet, outperforming other state-of-the-art methods under manageable compute and storage requirements. By analyzing data support estimates for each test sample, we find our approach can find smaller support sets in many cases. By trasferring attribution scores from one model to another, we investigate the importance of architecture in existing attribution methods. Additionally, our results highlight that the training data itself plays an important role in estimating data brittleness, providing valuable insights into data attribution.

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

# A APPENDIX

## A.1 COMPUTE TIME AND STORAGE REQUIREMENTS

For our compute time estimates, we use NVIDIA RTX A6000 GPUs and 4 CPU cores. We describe how we estimate the wall-clock time, and storage requirements for each method below -

- **Datamodels:** We only take into account the storage and compute cost of training models. The additional cost of estimating datamodels from the trained models, requires solving linear regression whose computational costs are negligible compared to training the models. For compute and storage requirement estimates, we train 100 ResNet-9 models on random $50\%$ subsets of CIFAR-10 and extrapolate to estimate the training time and storage required for 10,000 and 50,000 models shown in fig. 1.

- **TRAK:** We use the authors' original code [4] to train, and compute the projected gradients for CIFAR-10 using ResNet-9 Models using a projection dimension of 20480. For storage requirements, we take into account storage used by model weights, and the projected gradients. The results in fig. 1, show the compute and storage using 10, 20 and 100 models.

- **Ours:** We use Lightly library [5] benchmark code to train a MoCo model using a ResNet-18 backbone on CIFAR-10 for 800 epochs. The results in fig. 1 show the wall-clock training time for the model, and extracting the features from CIFAR-10 and the storage requirements for model weights.

To calculate the storage requirements, we factor in the storage space necessary for retaining the trained model weights, as they are essential for computing influence on new validation samples across all attribution methods.

## A.2 TRAINING SETUP

For CIFAR-10 (Krizhevsky et al., 2009), we train ResNet-9 [6] and MobileNetV2 (Sandler et al., 2018) models for 24 epochs using a batch size of 512, momentum of 0.9, label smoothing of 0.1, with a cyclic learning schedule, with a maximum value of 0.5. The test accuracy for these models without any modification to training data is above $92\%$. We randomly selected 100 validation samples, in a class-balanced manner for our brittleness metrics. We remove or mislabel a maximum of 1280 training samples for each validation sample. Our training setup is similar to Ilyas et al. (2022).

For ImageNet (Deng et al., 2009), we train ResNet-18 (He et al., 2015) models for 16 epochs, using a batch size of 1024. We train on $160\times160$ resolution images for the first 11 epochs and increase the training resolution to $192\times192$ for the last 5 epochs. The other hyperparameters are kept the same as CIFAR-10. These models achieve a top-1 validation accuracy of $67\%$. We randomly selected 30 validation samples, from a subset of validation samples that are not misclassified by 4 ResNet-18 models on average. We removed or mislabeled a maximum of 1000 training samples for each validation sample.

## A.3 ADDITIONAL SELF-SUPERVISED FEATURES

In addition to utilizing features from MoCo in section 3, we test our choice of distance function on ResNet-18 features from other self-supervised learning (SSL) methods trained on CIFAR-10. In particular, we evaluate BYOL Grill et al. (2020), SimCLR Chen et al. (2020), and DINO Caron et al. (2021) at estimating data removal support in fig. 8 and mislabel support in fig. 9. With the exception of DINO, self-supervised features from BYOL and SimCLR outperform the supervised baseline at estimating data removal support. Additionally, we see that in all cases using ESVM distance is more effective than using $\ell_2$ distance to compare features.

---

[4] https://github.com/MadryLab/trak

[5] https://github.com/lightly-ai/lightly

[6] https://github.com/wbaek/torchskeleton/blob/master/bin/dawnbench/cifar10.py

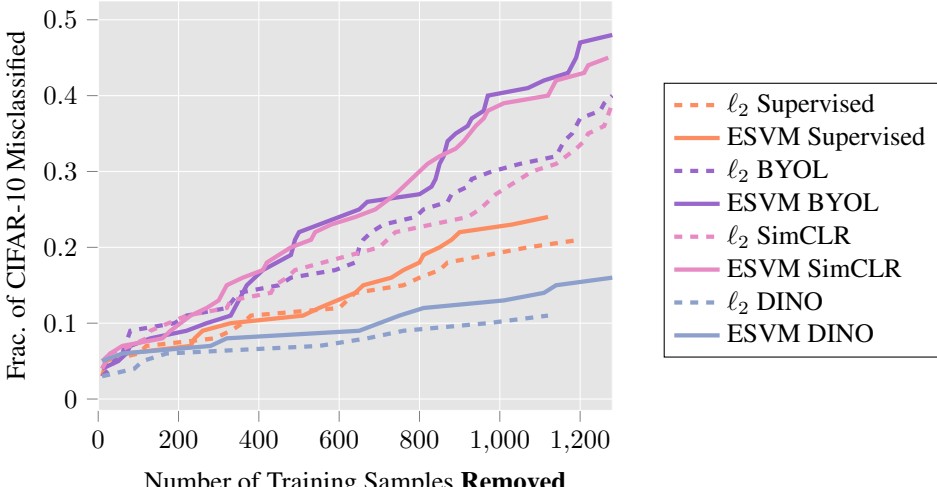

Figure 8: We estimate data removal support for 100 random CIFAR-10 test samples and plot the CDF of estimates.

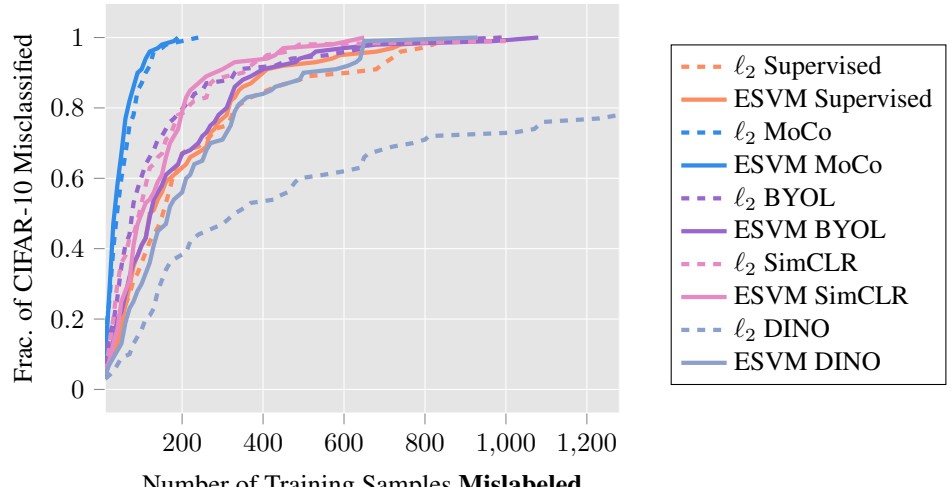

Figure 9: We estimate data mislabel support for 100 random CIFAR-10 test samples and plot the CDF of estimates.

### A.3.1 SELF-SUPERVISED IMAGENET FEATURES

We also consider using ImageNet features from MoCo v3 Chen et al. and DINO Caron et al. (2021) to estimate data removal support in fig. 10. We use publicly available MoCo v3 and DINO checkpoints from the `vissl` library's model zoo Goyal et al. (2021). It is worth noting that this approach places significant emphasis our primary hypothesis, which asserts the importance of visual similarity in data attribution. Utilizing ImageNet features means that the dataset, architecture, and learning objectives are *completely different* from the system we are trying to attribute predictions for: a ResNet-9 trained normally on CIFAR-10. This is in contrast to our main method (ESVM MoCo) which utilizes a ResNet-18 architecture and the CIFAR-10 dataset.

### A.4 ADDITIONAL JUSTIFICATION FOR CHOSEN SUBSET OF TRAIN IMAGES

For a target sample $z_t$, data attribution approaches rank the training samples based on decreasing order of positive influence on $z_t$. For our method, a design choice was whether to rank training samples from all classes or from a selected subset of the training data. One reasonable subset was

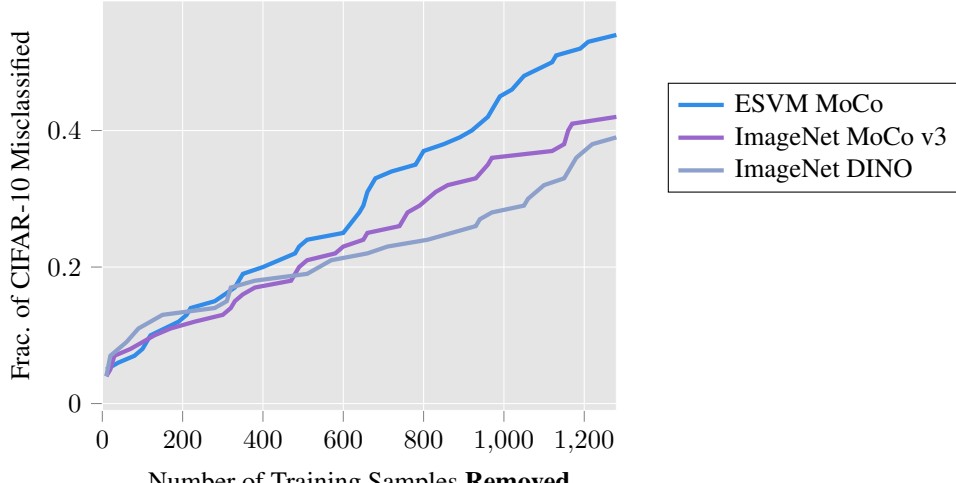

Figure 10: ImageNet features from MoCo v3 and DINO are able to perform very well despite using different architecture (i.e. ViT), dataset (i.e. ImageNet), and learning objectives (i.e. SSL) from the system that we are trying to attribute predictions for: a ResNet-9 trained normally on CIFAR-10.

to select training samples from the same class as the target test sample. In fig. 11, we show that selecting from the same class is more effective when estimating britteness scores. We maintain this choice for all our experiments.

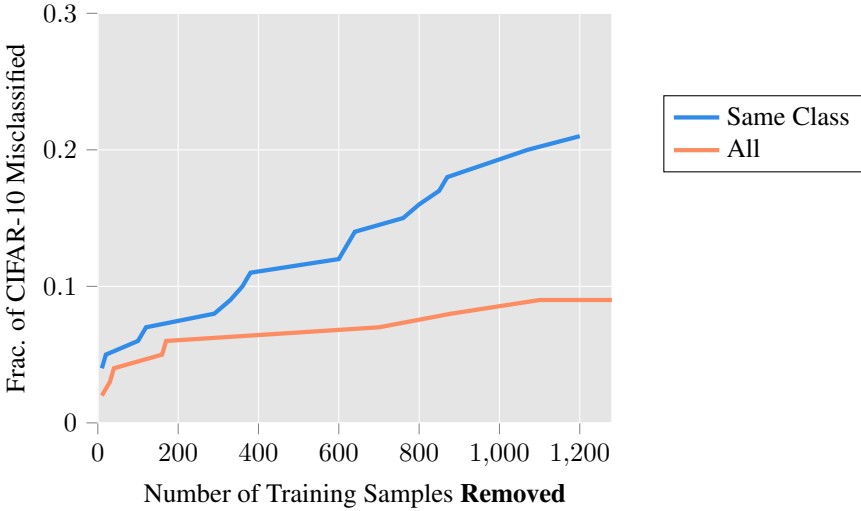

Figure 11: Choosing removal support from all training images is less effective than selecting from the same class as the target image.

## A.5 ADDITIONAL JUSTIFICATION FOR DISTANCE FUNCTION

In section 3, we describe choices for measuring similarity of embeddings: Euclidean distance, cosine distance, and our selection of Exemplar SVM. However, there are a range of other metrics that have been evaluated in prior work. By no means have we exhausted the space of possible metrics, but it relevant to look at recommendations by related work.

### A.5.1 GRADIENT COSINE SIMILARITY

Hanawa et al. (2021) define a set of tests that a similarity metric should satisfy and find that gradient cosine similarity (Grad-Cos) is the only one that passes all tests. Given that Grad-Cos is their overall recommendation for measuring similarity, we evaluate data removal support on CIFAR-10 in fig. 12. Note that unlike other methods considered, we do not filter images to be of the same class as the target because Grad-Cos already provides a higher ranking to images from the same target class. While we find that Grad-Cos is better than ESVM comparison of supervised features, it still lags behind our main method (ESVM MoCo) from section 3. Interestingly, in the low data support regime, where fewer than 200 training samples can be removed to misclassify, Grad-Cos is more effective than ESVM MoCo.

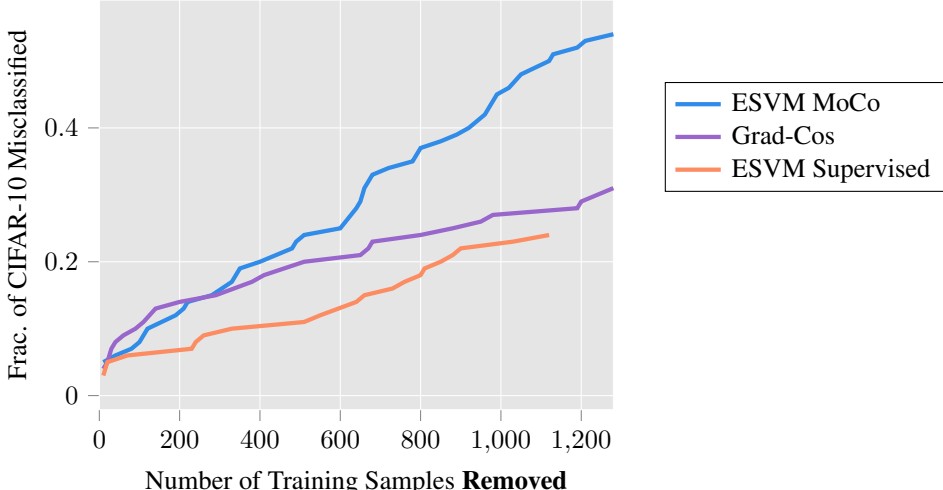

Figure 12: While comparing images with Gradient Cosine Similarity (using a supervised ResNet-9) is better than ESVM on supervised features, it still lags behind our main method (ESVM MoCo).

### A.5.2 HUMAN VISUAL SIMILARITY & DREAMSIM

Fu et al. (2023) study perceptual metrics and find that large vision models like OpenCLIP Cherti et al. (2023) and DINO Caron et al. (2021) are more aligned with human perceptual judgements than other learned metrics like LPIPS Zhang et al. (2018) and DISTS Ding et al. (2020). They further improve performance of OpenCLIP and DINO by finetuning with LoRA Hu et al. (2021) on a dataset of human two-alternative forced choice (2AFC) judgments, called NIGHTS. The best approach on the dataset uses an ensemble of DINO, CLIP, and OpenCLIP features and is called DreamSim. While the ensemble gets $96.2\%$ accuracy on NIGHTS, only utilizing OpenCLIP (with LoRA) gets $95.5\%$ and is $3\times$ faster. Hence, we use this metric in our data removal support evaluation. Here, we also select images from the same training class. In fig. 13, for every target image, we select the closest training images according to DreamSim to remove. Surprisingly, DreamSim does not improve over our approach using ESVM MoCo.

### A.6 COMPUTING DATA SUPPORT

We use bisection search to estimate data support. The use of bisection search is supported by the observation that several data attribution approaches are additive Park et al. (2023), where the importance of a subset of training samples is defined as the sum of each of the samples in the subset. To compute data removal support, we remove $M$ samples (chosen using each attribution method) from the training data and log whether the resulting model misclassifies the target sample. For data mislabeling support, we mislabel $M$ samples (chosen using each attribution method) from the training data and assign a new label corresponding to the highest incorrect logit.

A detailed summary of our bisection search is in algorithm 1. A key step is CounterfactualTest($f, S, I_{\text{attr}}[: M]$) which returns the average classification of $N_{\text{test}}$ indepen-

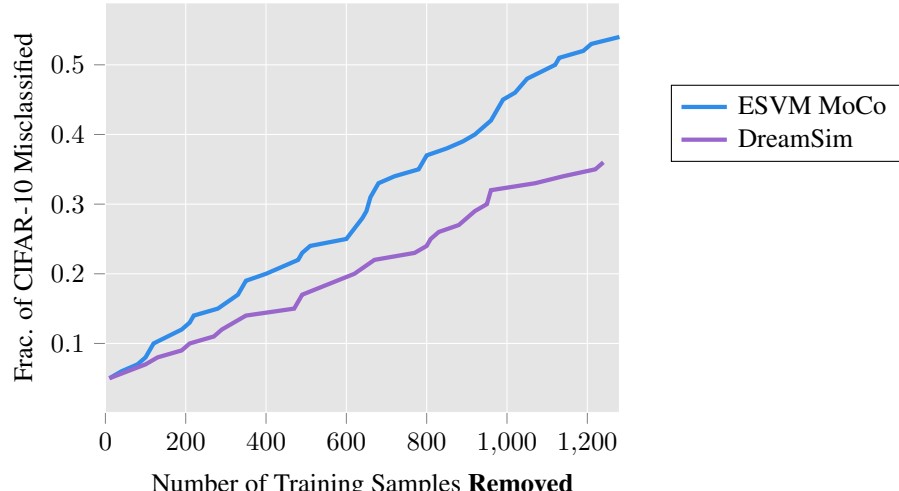

Figure 13: We use DreamSim (OpenCLIP-ViTB/32) to select data removal support on CIFAR-10.

dent training runs where $f_\theta$ is trained on the subset $R = \{z_i | z_i \in S \text{ and } i \notin I_{\text{attr}}[: M]\}$. In other words, for computing data removal support, $f_\theta$ is trained on a subset of $S$ that does not include the first $M$ indices of $I_{\text{attr}}$. For computing mislabeling data support, the only difference is that rather than removing the first $M$ indices of $I_{\text{attr}}$, we relabel those samples with the class of the highest incorrect-class logit, following Ilyas et al. (2022).

---

**Algorithm 1** Bisection Search for Computing Data Support

---

**Input:** Target sample, $z_t = (x_t, y_t)$
**Input:** Training set, $S$, and a list of top $k$ training set indices $I_{\text{attr}}$ ordered by the attribution method $\tau(z, S)$
**Input:** Model $f_\theta$
**Input:** Search budget, $N_{\text{budget}}$
**Input:** Number of times to test classification, $N_{\text{test}}$
**Output:** $N_{\text{support}}$, size of the smallest training subset $R \subset S$ such that $f_\theta$ misclassifies $x_t$ on average

1: $L \leftarrow 0$
2: $H \leftarrow |I_{\text{attr}}|$
3: $M \leftarrow H$
4: $C_{\text{avg}} \leftarrow \text{CounterfactualTest}(f, S, I_{\text{attr}}[: M])$
5: **if** $C_{\text{avg}} > 0.5$ **then**
6:     **return** -1                                   $\triangleright$ $N_{\text{support}}$ is larger than $k$
7: **end if**
8: $N_{\text{support}} \leftarrow M$
9: **while** $N_{\text{budget}} > 0$ **do**
10:     $N_{\text{budget}} \leftarrow N_{\text{budget}} - 1$
11:     $M \leftarrow (L + H)/2$
12:     $C_{\text{avg}} \leftarrow \text{CounterfactualTest}(f, S, I_{\text{attr}}[: M])$
13:     **if** $C_{\text{avg}} > 0.5$ **then**
14:         $L \leftarrow M$
15:     **else**
16:         $H \leftarrow M$
17:         $N_{\text{support}} \leftarrow \min(M, N_{\text{support}})$
18:     **end if**
19: **end while**
20: **return** $N_{\text{support}}$

---

Table 2: **Our baseline approach transfers well across diverse architectures, suggesting the role of data in counterfactual estimation is large**. We evaluate AUC for data removal support when transferring attribution scores to different architectures. We use attribution scores for TRAK and DataModels using the ResNet-9 backbone. For our approach, we use the ResNet-18 backbone. The number of models used by each method is shown in parentheses.

| | Ours (1) | TRAK (100) | Datamodels (50K) | Datamodels (300K) |
|---|---|---|---|---|
| ResNet-9 | 0.313 | 0.355 | 0.381 | **0.708** |
| ViT | **0.478** | 0.251 | 0.300 | 0.474 |
| SwinT | **0.517** | 0.347 | 0.303 | 0.495 |
| MLP-Mixer | **0.506** | 0.315 | 0.304 | 0.455 |
| ConvMixer | 0.324 | 0.302 | 0.337 | **0.645** |

For bisection search across all attribution methods, we use a search budget of 7. For the CIFAR-10 data brittleness metrics, we aggregate predictions over 5 independently trained models. Thus, to evaluate a single validation sample, we train 35 models (7 budget × 5 models) for a total of 3500 (35 × 100 samples) models for a data brittleness metric. On ImageNet, we don't aggregate predictions and only train a single model. Hence, to evaluate a single validation sample on Imagenet, we train 7 models per sample, and a total of 210 models for evaluating a data brittleness metric. Due to the large training cost on ImageNet, we only show results for data removal support. We explicitly point out that these costs are incurred only for analysis of these data attribution methods (see section 2). Our attribution approach is in comparison, extremely cheap to compute.

## B    TRANSFER TO OTHER DIVERSE ARCHITECTURES

In Table 2, we perform additional experiments on CIFAR-10, transferring the attribution scores to diverse architectures. We report the AUC for data removal estimates. Data removal estimation is computationally intensive and can require training thousands of models. Finding fast-to-train, performant architectures on CIFAR-10 that aren't convolution-based is difficult. We use the same evaluation setup, as Section 4.4 and modify the number of epochs to 50 for all architectures except ConvMixer. We also change the learning rate schedule for all models except ConvMixer for faster convergence. ConvMixer gets over 90% accuracy, and other models get 80-85% test accuracy with the same setup. We don't include results on MLP-based models, since the architectures we tried, achieved less than 70% CIFAR-10 test accuracy. Our proposed approach performs well across diverse architectures, outperforming all approaches including Datamodels with 300K models on ViT, Swin-Transformers, and MLPMixer. On ConvMixer, our approach performs nearly as well as Datamodels (50K). These results further suggest there exists an inherent form of data brittleness, independent of the model architecture and our proposed method can identify it effectively.

