# OpenReview forum: "Data Brittleness Estimation with Self-Supervised Features"
_ICLR.cc/2025/Conference — ICLR 2025 Conference Withdrawn Submission_

### Official Review · Reviewer_4Sfz · 2024-10-28

**Soundness:** 3
**Presentation:** 2
**Contribution:** 3
**Rating:** 3
**Confidence:** 3

**Summary:**

This paper proposes a new method to enhance computational efficiency in estimating data brittleness. While previous models used up to 300k models to measure data sensitivity, this paper leverages a single self-supervised model feature. For measuring data brittleness, two methods are employed: Data Removal Support, which identifies the minimal subset of training data that, when removed, degrades model performance, and Data Mislabel Support, which examines the effects of manipulated labels in the training data. Experiments were conducted on CIFAR-10 and ImageNet, showing performance comparable to or better than Datamodels (300K models) or TRAK (100 models) in certain cases. Additionally, this paper demonstrates that the conventional approach, which focused solely on the Linear Datamodeling Score (LDS), does not necessarily correlate with data brittleness measurements.

**Strengths:**

1. The target task in this paper requires an approach focused on computational efficiency, and the authors aim to address this by performing estimation using only a single self-supervised model feature.
2. In the process of measuring data brittleness, the approach is not dependent on specific model architectures or training processes, thus enhancing the focus on the intrinsic characteristics of the data itself.

**Weaknesses:**

1. The paper does not clarity problem definition/formulation. Since the model's inherent characteristics (bias, etc.) could impact the evaluation depending on how the problem is defined, a proper problem definition is necessary to justify the use of only a self-supervised backbone in this paper.
2. The paper showed very low performance on the LDS metric. Although Section 2.1 critiques LDS and other existing metrics, the paper does not propose an alternative quantitative metric. Line graphs and histograms only provide qualitative insights and cannot serve as quantitative metrics.
3. Performance was validated only on the CIFAR-10 and ImageNet datasets, raising uncertainty about whether this approach can be generalized more broadly.

**Questions:**

Please refer to the weaknesses.

---

### Official Review · Reviewer_fMuZ · 2024-10-30

**Soundness:** 2
**Presentation:** 2
**Contribution:** 2
**Rating:** 3
**Confidence:** 3

**Summary:**

This paper proposes a method to measure the contribution (brittleness) of training data samples to the accurate prediction of a given test image by a vision classifier. Unlike existing methods that require large storage and computation due to numerous models or embeddings, the proposed method is highly efficient, requiring only a single model trained through self-supervised learning. Specifically, the method estimates data brittleness by measuring the distance between the target sample and training samples within the space of the pre-trained model. The distance function used includes Euclidean distance and ESVM-based distance functions. The authors demonstrate that models trained with supervised learning are less effective at estimating data brittleness and, through experiments on CIFAR-10 and ImageNet, show that the proposed method is both more efficient and effective than existing approaches.

**Strengths:**

1. The proposed method is lightweight and easy to implement.
2. The proposed method outperforms existing methods in the tested settings.

**Weaknesses:**

1. There is insufficient explanation for the figures inserted in the text:
    1. Each figure is not referenced in the main text.
    2. In the right graph of Figure 1, what search algorithm was used for each method?
2. The importance of the problem being addressed in this paper and how it can be applied in real-world scenarios are not discussed, making the advantages and disadvantages of the proposed method less prominent. For example, since brittleness metrics only account for positive influence in training samples, it can be expected that the method will not be effective in filtering out noisy data, such as incorrectly annotated samples.
3. It is necessary to verify the effectiveness of the method in domains other than natural domains like CIFAR-10 and ImageNet.
4. By using pre-trained models, the brittleness of data not actually used in training the classifier may be estimated as very high.

**Questions:**

1. In the right graph of Figure 1, what search algorithm was used for each method?
2. Why is the problem being addressed important, and how can the proposed method be applied in real-world scenarios?
3. How effective is the proposed method in domains other than natural domains like CIFAR-10 and ImageNet?
4. By using pre-trained models, the brittleness of data not actually used in training the classifier may be estimated as very high. What are your thoughts on this issue?

---

### Official Review · Reviewer_McZb · 2024-11-03

**Soundness:** 2
**Presentation:** 3
**Contribution:** 3
**Rating:** 5
**Confidence:** 2

**Summary:**

The paper makes a contribution to the “data attribution” literature, an area that aims to trace model prediction back to the training data.
More specifically, this paper introduces a method for estimating "data brittleness," which is the sensitivity of model predictions to changes in the training data. The authors propose a computationally efficient baseline approach using image features from a pre-trained self-supervised backbone. Their method contrasts with state-of-the-art data attribution techniques that require ensembles of models and intensive computational resources. They validate their approach on CIFAR-10 and ImageNet datasets, showing it can outperform traditional attribution methods in computational efficiency and data brittleness estimation accuracy. The work aims to highlights a potential new application for self-supervised pre-training and a new angle on data brittleness.

**Strengths:**

1. The work is clearly motivated, highlighting the high computational costs of current approaches.
2. The method is more efficient and scalable compared to previous approaches and is benchmarked against those.
3. The paper is the first to connect self-supervised pre-training with data attribution.

**Weaknesses:**

1. The method’s practical applications are limited as it cannot be used to assess model bias/fairness as stated by the authors.
2. While the improved efficiency is obvious and well demonstrated, questions about the interpretability of this method’s metrics remain unclear (see Q1 and Q2 in particular). I am willing to reconsider my rating if the authors can provide satisfactory answers to these questions.
3. The work highlights the importance of image similarity in data attribution. However, the meaning of ‘image similarity’ can be different depending on the dataset at hand. CIFAR-10 and ImageNet classes are distinguished by coarse-grained features. It remains unclear if the results hold for more fine-grained datasets (e.g., CUB or iNaturalist) where class differences are more nuanced.

**Questions:**

1. If I understand you correctly you see data brittleness metrics as model-agnostic assuming that any classification model in some way makes predictions based on visual similarity. If this is the case, I am wondering if it is possible to measure a model-agnostic metric using the embeddings of a single model only.
2. Related to the previous question: You claim that your method “outperforms” baselines as you find a smaller subset of training samples that - when removed or relabelled - flip test set predictions. I would argue that the measured effect can be partly attributed to the dataset and partly to the model you are generating the embeddings with. In other words, if your model robustness is low, it will be easier to flip predictions. How can you then ensure a fair comparison to other data attribution methods that use different models or an ensemble of those?
3. I am surprised that DINO does worse than supervised features on CIFAR-10 (as stated in L 220), do you have a hypothesis why this is the case?
4. Fig 2: The empirical finding that using a linear ESVM is a distance measure than L2 is interesting. It seems to be the case in particular for large data removal support while the two metrics are similar when only a few training samples are removed. Do you have an interpretation for this?
5. Fig 3: Your approach does much better on the “Mislabel” metric than on the “Removal” metric compared to the baselines. I am missing a discussion on this particular result. Can you maybe share your thoughts on this?

**Minor comments:**
- The correct use of author-year citations would improve readability. I.e.: Author (2024) for in-text citations and (Author, 2024) elsewise.
- L. 259: missing period at end of paragraph
- Missing periods/colons for paragraph titles in lines 228 and 239
- Fig. 1 there seems to be a text artifact (“t”) in the left panel
- L. 833: Typo in “brittleness”

---

### Note · Authors · 2024-12-05

I have read and agree with the venue's withdrawal policy on behalf of myself and my co-authors.